# Assessment of the role of anthocyanin standardized elderberry (*Sambucus nigra*) extract as an immune-stimulating nutraceutical of Nile tilapia, *Oreochromis niloticus*

**Md Mursalin Khan¤a, Asif Mortuza¤b, Md Ibrahim¤c, Ahmed Mustafa** *

Department of Biology, Purdue University Fort Wayne, Fort Wayne, IN, United States of America

¤a Current address: Department of Biological Sciences, Auburn University, Auburn, AL 36849, United States of America
¤b Current address: Department of Marine Biology, Texas A&M University at Galveston, Galveston, TX, United States of America
¤c Current address: Department of Environmental Science, Baylor University, Waco, TX, United States of America
* mustafaa@pfw.edu

**Data Availability Statement:** All relevant data are within the paper.

## Abstract

The study of nutraceuticals and their connection to immunity is an expanding field of research. The use of nutraceuticals to alleviate stress and enhance immunity in adverse aquaculture environments have been examined to a certain extent. To elucidate the understanding, we focused on the immunological effect of membrane-separated 13% anthocyanin standardized elderberry (EB) extract with maltodextrin excipient, widely used first-line nutraceuticals to augment the immunity, in aquaculture fish, Nile tilapia. To evaluate the potential of EB-extract, we assessed their capability to enhance lymphocyte proliferation and interleukin-2 production in an in-vitro condition using spleen and thymus lymphocytes. The experiments on spleen and thymus T-cells demonstrated significantly higher T-cell proliferation by EB-extract when lectin mitogen Con A was present as a stimulator. Likewise, our spleen B-cell proliferation result reveals a significant effect of EB-extracts, along with B-cell stimulator non-lectin mitogen LPS. Further, the quantification of IL-2 indicates elevated IL-2 levels when spleen T-cells were cultured with EB-extracts and with Con A present as a stimulator. These suggest that 13% anthocyanin standardized EB-extracts can aggrandize fish cells' cellular and humoral immune responses. With further research, elderberry extracts could be used to supplement commercial feed in aquaculture to reduce stress and stimulate the immune response.

## 1. Introduction

Natural food or nutraceutical-mediated solution to alleviate diseases, lessen stress and enhance the immune system is one of the fastest-growing fields of the present time [1–3]. Due to the

**Funding:** The author(s) received no specific funding for this work.

**Competing interests:** The authors have declared that no competing interests exist.

various deleterious effect of chemical drugs and antibiotics, scientists are trying to use natural sources to prevent or treat various diseases [2, 4–7]. The natural food-based remedy for diseases via enhancing immune response has been reported in many biological systems such as humans, rats, mice, and fish [8–11]. Among them, arguably, darkly pigmented fruits are attracting the highest attention due to their diverse health benefits and wide availability [3, 12]. Recent studies showed that darkly pigmented fruits such as berry extract could decelerate the aging process, alleviate hypertension and cardiovascular diseases, stress modulation, and improve protection against infectious agents [1–3, 13, 14]. Although several attempts have been made by researchers to distinguish the role of various nutraceuticals in different model systems [2, 15–17], the comprehensive insights of the nutraceutical are yet to be uncovered due to inadequate studies in various models, limited knowledge about the potential effect of processing techniques specifically extract preparation and enrichment, and lack of data regarding bioavailability and membrane interaction [2, 18–20]. Thus, to explicate the specific role of nutraceuticals on immune response, we investigated the effect of 13% anthocyanin standardized Austrian black elderberry (*Sambucus nigra* 'Haschberg') extract in cell-mediated and humoral immunity in *in-vitro* conditions using Nile tilapia, *Oreochromis niloticus*.

The predominant goal of our research is to observe the ability of 13% anthocyanin standardized Austrian black elderberry (EB) extract in stimulating lymphocytes for proliferation using the spleen and thymus of Nile tilapia; additionally, the researchers noted the impact of EB extract on cytokine release, in particular, interleukin-2 (IL-2). In this study, we relied on lymphocyte proliferation assay to detect the effects of elderberry on spleen and thymus cells and measurement of the cytokine release (IL-2) by T-helper cells. We are particularly interested in T-cells as it has the capability to detect and remove cells that are infected by viruses or are cancerous [21]. Furthermore, T-cell lymphocytes can send signals to other cells to fight against infected cells and thus play a role in assisting innate and adaptive immune responses [21]. T-cells originate from stem cells in the bone marrow and travel to the thymus to become mature; thereupon, matured T-cells migrate to the spleen and other lymphoid tissues [21]. The thymus is the place where they mature and learn to respond to non-self; in consequence, the thymus supplies the educated T-cells to the spleen and lymph nodes to carry out their immune functions [21]. Thus, spleen T-cells are ready to combat antigens and can go through cell activation and proliferation [21]. Therefore, there is a significant difference between the thymus and spleen T-cell population in the corresponding organs, which is worth noting while observing the impact of EB extracts. In this study, we stimulated both spleen and thymus T-cells using mitogen Concanavalin A (Con A) to examine the impact of the EB extract on these two vital immunogenic organs [19, 22].

Con A is a plant-derived lectin, a carbohydrate-binding protein that recognizes specific carbohydrate moieties on membrane glycoproteins on the surface of the cells [23]. In the T lymphocytes, the binding of certain lectins such as Con A to carbohydrates of T cell Receptors (TCR) or CD3 chains is adequate to activate intercellular signaling cascade, which results in subsequent T cell activation and proliferation [23]. As the carbohydrate elements are often the same regardless of the antigenic specificity of the TCR, mitogen Con A can activate many T cell clones of various specificities and thus functions as a polyclonal activator or stimulator [23, 24]. This study analyzed the effect of elderberry extract to increase the Con A specific polyclonal proliferation of the T-cells for both the spleen and thymus tissues. As Concanavalin A (Con A) in the solution has been reported to induce T cell proliferation but not B cells, in our experiment, it is anticipated that only T-cells will proliferate [24].

On the other hand, B-cells are the most important part of humoral immunity [21]. The B-cells produce antibodies used to eradicate deleterious bacteria, viruses, and toxins [21]. B lymphocytes recognize antigens and produce specific antibodies to eliminate harmful elements

[21]. After generating in the bone marrow, B-cells migrate to the spleen and other secondary lymphoid tissues, where they mature and differentiate into immunocompetent B-cells[21]. B-cells make antibodies to specific antigens, which bind to antigens via B-cell receptors [21]. In our study, B-cells were obtained from the spleen to see the effect of the 13% anthocyanin standardized EB extracts and mitogen lipopolysaccharide (LPS) on humoral immunity.

LPS has been widely used as a highly mitogenic substance to stimulate B-cell proliferation upon activation [19, 25]. LPS is a non-lectin mitogen for B cells; it binds not to the B cells receptors (BCR) but CD14 and TLR4 on the membrane of the B cells, leading to a cascade of cellular signaling, and eventually activation and proliferation of the B-cell lymphocytes [26]. Although LPS stimulation predominantly leads to B cells proliferation in the lymphoid cells, there are several findings that showed that LPS could stimulate T cells *in vivo* [27–29]. Therefore, it is well established that LPS can act as an adjuvant for T-cell responses to specific antigens *in-vivo* [30]. Our study analyzed the effect of EB extract in increasing the LPS-specific clonal proliferation of B-cells; as there is no specific antigen present, we presume there is no proliferation of *in-vitro* T-cells.

Communication among the immune cells is fundamental for the success of both cellular and humoral immune responses [31, 32]. Cytokines are one of the essential signaling molecules that communicate among immune cells to generate a highly interconnected network [31, 32]. IL-2 is one of the most important cytokines mostly released by T-helper cells (activated CD4 T cells), with other IL-2 producers being cytotoxic T cells (CD8 T cells), DCs, NKT cells, and mast cells [21, 33, 34]. IL-2 commutes to other immune cells to transmit the signals for a coordinated immune response [33, 34]. IL-2 can bind to a heterotrimeric protein expressed on the surface of certain immune cells called interleukin-2 receptor (IL-2R) and provide signals [31, 33, 34]. IL-2 can induce the proliferation of T-helper cells, cytotoxic T-cells, and NK-cells [35, 36]. Moreover, IL-2 regulates immune stimulation by binding to T-Regulatory cells (T-Reg) with higher affinity; eventually, this high-affinity binding leads to a negative feedback mechanism to decrease CD8 T-cell activity through both depletion of IL-2 and inhibition by T-Reg cells [34–36]. T helper 1 cells are mostly involved in cellular immunity to eliminate intracellular pathogens and cancer cells; T helper 2 cells engage in humoral immunity via promoting antibody production by B-cells to combat extracellular pathogens [32]. Our experiment measured the IL-2 production rate of the spleen T-cells (mostly activated CD4 T cells) in the cell culture media in the presence of elderberry extracts. Con A was used as the mitogenic agent for stimulating the T-helper cells (activated CD4 T cells), not unlike in the case of the T-cell proliferation assay [22]. Enzyme-linked immunosorbent assay (ELISA) was used to quantify the IL-2 level of T-helper cells in the presence of elderberry.

To examine the effect of nutraceuticals on the immune cells of the Nile tilapia, we used the extract of a dark pigmented fruit, elderberry, well known for its potential nutritional and therapeutic effects [37]. In the context of the Covid-19 pandemic, attention toward elderberry extracts increased considerably because of the particular interest in elderberry's antiviral effects [3, 12, 20, 38]. 13% anthocyanin standardized elderberry (*Sambucus nigra*) extracts contain several bioactive components such as anthocyanins, polyphenols, phenolic acid, flavonoids, catechins, and gallic acid [39–43]. The primary anthocyanins of EB extracts are cyanidin-3-glucoside and cyanidin-3-sambubioside, which can activate apoptotic pathways, arrest cell cycles, interact with lipid membranes, and inhibit tumor-associated enzyme secretion [18, 41, 42, 44]. Phenolic compounds can be classified into several groups, such are phenolic acids, flavonoids, tannins, coumarins, lignans, stilbenes, and others, as well as into different subgroups [2, 41]. The phenolic compounds of dark-pigmented berries have been reported to reduce cancer cell growth [13, 41]. Berry's antioxidants react with reactive oxygen species (ROS) and neutralize them, thus preventing damage to cellular components such as DNA,

lipids, and proteins [13, 45, 46]. In addition, berries have important flavonoids, and anthocyanins, which can help maintain homeostasis of cellular functions and suppression of melanoma cell growth [1, 44]. Most of these secondary plant metabolites exert *in vivo and in vitro* antioxidant activity [8–10], and have been observed in rats [11, 47] and humans [19, 43, 48].

Due to the wide range of health benefits that can be achieved using various berries such as elderberry (EB), it is important to assess their effect on various species, for instance, fish and shellfish. However, there is very little is known about the effect of elderberry on fish models [14]. In our study, we are using one of the most cultivated aquaculture fish, Nile tilapia, to decipher the role of elderberry extract on the fish's immune system. Nile tilapia is a widely cultivated organism in aquaculture all around the world due to its superior protein quality and high growth capabilities even under intense aquaculture-related stresses such as crowding stress, competition for food, poor development, accumulation of deleterious metabolites in the water, higher occurrence of diseases, diminished survival, and cannibalism [49, 50]. If elderberry can be used as a supplement in the feed of the fish to mitigate stress and improve the immune system, it will be a healthier replacement to using chemical drugs and antibiotics, which can lead to pollution and affect non-target species. Hence, we examined the capability of elderberry (EB) extract in stimulating cell-mediated immunity in both Thymus and Spleen T-cell proliferation and humoral immunity in Spleen B-cell proliferation. In addition, we observed the ability of the EB extract in increasing the IL-2 production of spleen T-helper cells.

## 2. Methods and materials

### 2.1. Fish acquisition and maintenance

In the experiment, Nile tilapia (*Oreochromis niloticus*) was used as the model organism. Disease-free juvenile Nile tilapia were acquired from Troyer Farms, Indiana. The average weight and length of the fish were 35±1 g and 13±2 cm, correspondingly. The fish were kept under standard optimal conditions (pH: 6.0–7.0, ammonia: 1.0–3.0 mg/dL, temperature: 25–28°C, dissolved oxygen 5.00–7.00 mg/L) in a recirculating aquaculture system. They were acclimated to experimental tanks for four weeks prior to starting the experiments. The Nile tilapia were fed commercial 3/32" extruded pellet feed, Purina® AquaMax® Grower 400 once a day at 3% of their body weight. Commercial Food Composition: Crude protein 45%, Crude fat 16%, Crude fiber 3%, Sodium 0.60%, Calcium 1.7%, Phosphorous 1.2%.

### 2.2. Anthocyanin standardized elderberry extracts

The 13% anthocyanin standardized elderberry extracts used in the experiment is from Artemis International Inc (Fort Wayne, IN). This extract is a Austrian black elderberry (*Sambucus nigra*, L.) extract, which was processed by Artemis International Inc without any organic solvents using physical membrane separation and spray-dried with maltodextrin as an excipient. According to the manufacturer, the physical membrane separation technique separates the bioactive components of the elderberry in a solution by eliminating undesirable substances without using any organic solvent to keep the integrity of the elderberry. Then, spray drying technique was used in the manufacturing process to dry the extract into a powder. Spray drying also helped in avoiding any heat-dry damage to the bioactive components. During the spray drying process, organic corn-based maltodextrin was used as a drying aid for its role as a carrier or a condensing agent. The organic corn-based maltodextrin is an excipient like gelatin, an inactive substance serving as a medium in the extraction process. According to the literature accompanying the product, the elderberry extract was standardized to a minimum anthocyanin content of 13% with an extract ratio of 64:1 by the manufacturer. HPLC was used to detect the anthocyanin profile unique to elderberry. Nutritional Profile reported by the

manufacturer (Mean values per 100 g): Calories (kcal) 343 kcal, Fat 0.2g, Saturated Fat < 0.2g, Carbohydrates 76.4g, Sugars 23.8g, Protein 6.6g, and Salt 0.050g. Analytical Profile (per 100g): Anthocyanins (Expr. as cyanidin-3-glucoside, pH-Diff.) 14.00–18.00 g, Anthocyanins (Expr. as cyanidin-3-glucoside, HPLC) 12.50–17.50 g, Polyphenols (Expr. as catechin, Folin-Ciocalteu) 18–35 g, Polyphenols (Expr. as gallic acid equivalent; Folin-Ciocalteu) 16.6–33.2 g, Moisture 2.0–8.0%, Total acidity (Expr. as citric acid, anhydrous) 9.00–19.00 g, according to the literature accompanying the product.

### 2.3. Experimental design

To measure the T-cells proliferation, we used standardized stock concentration (1 mg/mL of $dH_2O$) of the black elderberry extract in three different concentrations (23.81 µg/mL, 46.51 µg/mL, and 88.88 µg/mL) in the presence of Con A (2.38 µg/mL) as the mitogen. We also used identical concentrations of elderberry extract in the presence of PAMPs (Pathogen associated molecular patterns) LPS (2.38 µg/mL) to observe the spleen B-cells proliferation. Additionally, we quantified the IL-2 production rate by spleen T-helper cells in the presence of Con A (2.38 µg/mL) using the same EB extract concentrations. The groups tested in the assays were: 1) Control: T/B-cells; 2) T/B-cells + mitogen (ConA/LPS); and 3) T/B-cells + mitogen (ConA/LPS) + elderberry extract at 23.81 µg/mL; 4) T/B-cells + mitogen (ConA/LPS) + elderberry extract at 46.51 µg/mL; 5) T/B-cells + mitogen (ConA/LPS) + elderberry extract at 88.88 µg/mL. All the samples were tested in triplicates, and the experiment was replicated.

### 2.4. Sampling

The fish were euthanized with Tricaine Methane Sulfonate (MS-222) to collect the spleen and thymus (Sigma-Aldrich; St. Louis, MO) at 200 mg/L. MS-222 was administered within 2 minutes of catching the fish to reduce stress from handling. Aseptic techniques were used for the removal of the spleen and thymus. The isolated spleen and thymus were kept in a 1x phosphate buffer solution (PBS) for the cell culture.

### 2.5. Cell culture

A single-cell suspension was made from the spleen and thymus for the cell culture. Isolated organs were placed on top of a heat-sterile sieve/mesh in a petri dish. The cells were suspended in 1ml of RPMI 1640 (Mediatech, Inc, Herndon, VA) by poking with a sterile plunger and collected in a non-adhesive syringe. A 5 µL of the single-cell suspension was counted in a hemocytometer by using 295 µL of isotonic solution (pH 7.0), 100 µL of lysis buffer (to remove red blood cells), and 100 µL of Trypan blue. Based on the cell count, the solution was standardized so that $1 \times 10^6$ cells were placed in each well of the 96 well plates with 100µl RPMI 1640 (with 10% Fetal Bovine Serum (FBS); Sigma, Cleveland, OH) and 10 ml/L Penicillin-streptomycin (MP Biomedicals, LLC, Ohio) to inhibit microbial growth.

### 2.6. Lymphocyte proliferation assay

For the T lymphocyte proliferation of the spleen and thymus, mitogen Con A was used. 0.5 µL of mitogen Con A (stock 100 µg/mL) was placed in each well of the 96 well plates. Thus, 2.38 µg/mL was the final concentration in each treatment well. The thymocytes and splenocytes were not fractionated to contain only T-cells, as ConA only stimulates T cells in *in-vitro* without specific antigens [24]. In the B lymphocyte proliferation assay of the spleen, B-cells were stimulated by the non-lectin mitogen LPS (final concentration of 2.38 µg/mL). The splenocytes were not sorted to contain only B-cells as LPS predominantly stimulates only B-cells

in the absence of any specific antigen in *in-vitro* [30]. For the T-lymphocyte proliferation of the spleen and thymus and B-lymphocyte proliferation assay of the spleen, the following procedures were followed.

The elderberry preparation used for the experimentation was 13% standardized *Sambucus nigra* elderberry powder, generously provided by Artemis International, Inc. (Fort Wayne, IN). 5 μL, 10 μL, and 20 μL of the elderberry extracts (stock concentration 1 mg/mL) were used; eventually, these led to a final concentration of 23.81 μg/mL, 46.51 μg/mL, and 88.88 μg/mL in the wells. With these treatments, the cells were cultured at 28˚C, in 5% $CO_2$ and 95% humidity for 48 hours. After 48 hours, the cells were monitored under an inverted microscope to see the proliferation and check for microbial contamination. Then 10μL (working concentration: 0.0375 μCi (Curie)/ μL) of the radioactive 3Thymidine (3H) (Moravek Biochemicals, Stock Concentration: 1mCi/mL) was added to each of the wells, making for a final concentration of 0.375 μCi. The cells were incubated in an identical condition for 24 hours. The proliferative cells uptake the radiolabeled 3Thymidine (3H) as a component of the chromosome while they are synthesizing DNA strands for mitotic cell division, therefore, is proportional to cell proliferation.

The DNA of the cells was harvested using a cell harvester (Brandel Cell Harvester: model# M-24). The cells in each well of the microtiter plate were harvested on filter paper (Whatman™, Buckinghamshire, UK) strips. 10% trichloroacetic acid (TCA) was passed through the cell harvester five times to precipitate the DNA. Ethanol was passed through to wash any free 3Thymidine (3H) off the filter paper discs. TCA solution was passed through the cells to rupture them. Then Ethanol was used to fix the DNA onto the filter papers. The filter papers were air-dried for 3 hours and then transferred to scintillation vials. 3 mL of scintillation cocktail (Ecolume™ scintillation cocktail, MP biomedicals, Southern CA, CA.) solution was added to each vial to submerge the filter paper. The vials were put into a scintillation counter (Beckman Coulter™ LS 6500 Multi-Purpose Scintillation Counter) to determine the radioactivity count. The emitted beta radiation per minute was counted as counts per minute (CPM) in the scintillation counter using a beta counter filter (Geiger counter).

## 2.7. Quantification of the lymphokine

The lymphokine Interleukin 2 (IL-2) is released from T-cells (mostly CD4+ cells, some by CD8+ cells). The spleen T-cells were cultured in the presence of Con A (2.38 μg/mL) in 12 well culture plates under the same culture conditions, and EB extract concentrations as the lymphocyte proliferation assay mentioned earlier in Section 2.6. To get an adequate amount of supernatant for the IL-2 assay, each factor's volume was set five times higher than previously described in Section 2.6. The incubation period for the IL-2 production by T-cells (mostly CD4+ cells, some by CD8+ cells) was 24 hours. After incubation at 28˚C in 5% $CO_2$ and 95% humidity, the supernatant was collected from each of the wells of the 12-well plate. The collected supernatant was then used to perform the Enzyme-Linked Immune Sorbent Assay (ELISA) following the protocol from Mouse IL-2 immunoassay kit (R&D user manual, Minneapolis, MN).

## 2.8. Statistical analysis

The data collected through the course of this experiment were analyzed using SigmaStat® 13.0, Systat Software Inc. For each of the assays, a one-way analysis of variance (ANOVA) was performed to determine whether the differences between the samples were significantly different (P * < 0.05, P ** < 0.01, P *** < 0.001). Subsequently, Tukey's test and Student's simple t-test were performed to determine which treatment groups were significantly different from

each other. The findings of our analyses are presented in the text and graphs in the form of means ± standard errors of means (SEM).

## 2.9. Ethical statements

All fish were taken care of following an approved protocol (PACUC Protocol Number: 112000481), approved by the Purdue University Animal Care and Usage Committee (PACUC) following the guidelines of the US National Research Council's Guide for the Care and Use of Laboratory Animal and Purdue University Aquatic Animal Standard Operating Procedures.

# 3. Results

## 3.1. Spleen T-cell proliferation assay

Spleen T-cell proliferation assay showed promising results in using 13% anthocyanin standardized elderberry extracts to stimulate the immune system by increasing the spleen T-cell proliferation rate. 13% anthocyanin standardized elderberry extract was used in three different concentrations along with mitogen Con A (2.38 μg/mL): 23.81 μg/mL, 46.51 μg/mL, and 88.88 μg/mL (Fig 1). Elderberry extract demonstrated a significantly higher T-cell proliferation rate (P < 0.001) than the control and ConA treatments. However, the rate of proliferation was

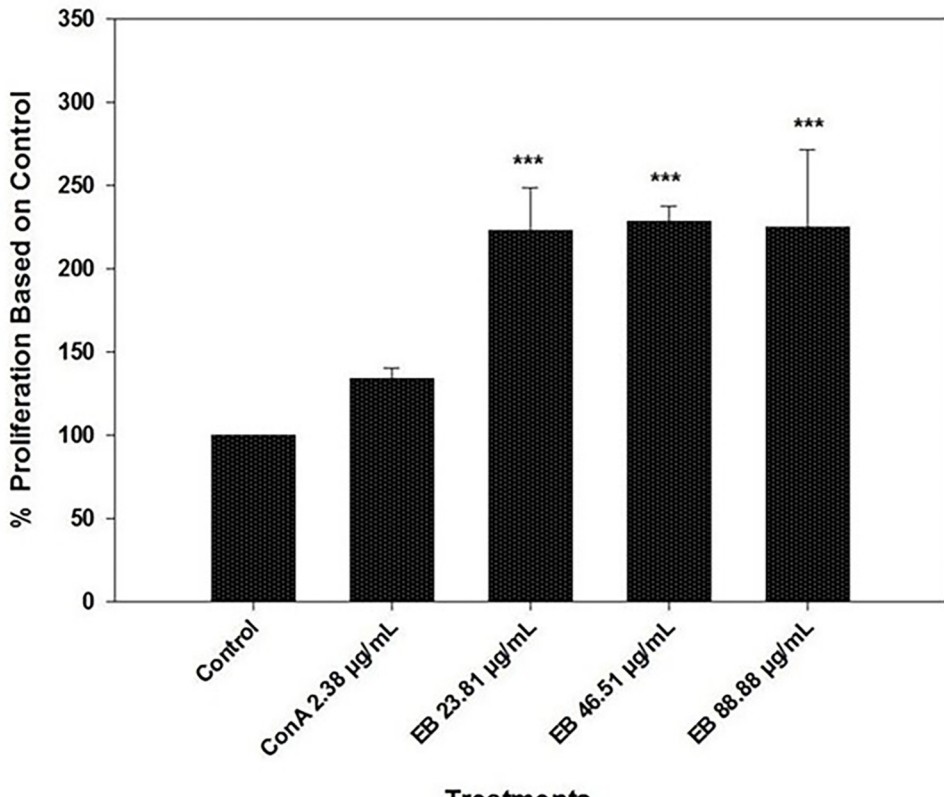

**Fig 1. Proliferation of fish spleen T-cells in the prescence of Con A and various concentrations of elderberry extracts (23.81 μg/mL, 46.51 μg/mL, and 88.88 μg/mL).** All the elderberry extract concentrations were tested in the presence of Con A. Results are represented in means ± SEM. Asterisks represent a significant proliferation difference between treatment and control cell growth (*** = P<0.001).

not increased in a concentration-dependent manner. In the 23.81 μg/mL, the proliferation was ~2.1 times higher than the control T-cell proliferation. The dosage increments (46.51 and 88.88 μg/mL) of the elderberry extract exhibited a similar level of T-cell proliferation compared to the control (P < 0.001) (Fig 1). Con A (2.38 μg/mL) was used to stimulate T-cells in the absence of elderberry extract, which also showed higher proliferation than the control, but it was not statistically significant (P > 0.05).

### 3.2. Thymus T-cell proliferation assay

The Thymus T-cells showed an analogous proliferation rate compared to the control at 23.81 μg/mL concentration (P > 0.05); moreover, at 23.81 μg/mL EB extract concentration, the proliferation was lower than ConA only (2.38 μg/mL) treatment (Fig 2). Nevertheless, the higher concentration of the elderberry extracts (46.51 and 88.88 μg/mL) showed a significantly higher rate of Thymus T-cell proliferation (Fig 2). The 46.51 μg/mL and 88.88 μg/mL concentrations were capable of stimulating T-cell proliferation significantly compared to control and ConA treatments (P < 0.001) (Fig 2). Contrary to the spleen T-cell proliferation, where there was no effect of higher concentration in spleen T-cell proliferation (Fig 1), the Thymus T-cell proliferation demonstrated a significantly higher proliferation rate at 88.88 μg/mL than 46.51 μg/mL treatments (Fig 2). In addition, the data represents a dose-dependent increase in T-cell proliferation to the concentration of the extract from 23.81 μg/mL to 88.88 μg/mL (Fig 2), which was not observed in the spleen T-cell proliferation assay (Fig 1).

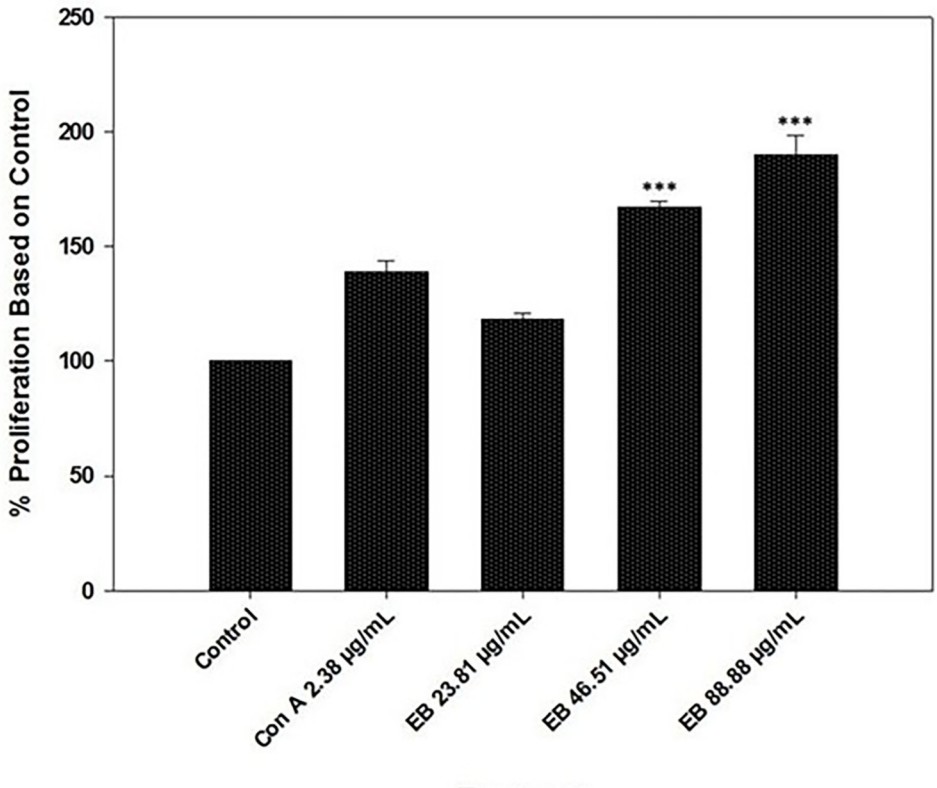

**Fig 2. Proliferation of fish thymus T-cells in the prescence of Con A and various concentrations of elderberry extracts (23.81 μg/mL, 46.51 μg/mL, and 88.88 μg/mL).** All the elderberry extract concentrations were tested in the presence of Con A. Results are represented in means ± SEM. Asterisks represent a significant proliferation difference between treatment and control cell growth (*** = P<0.001).

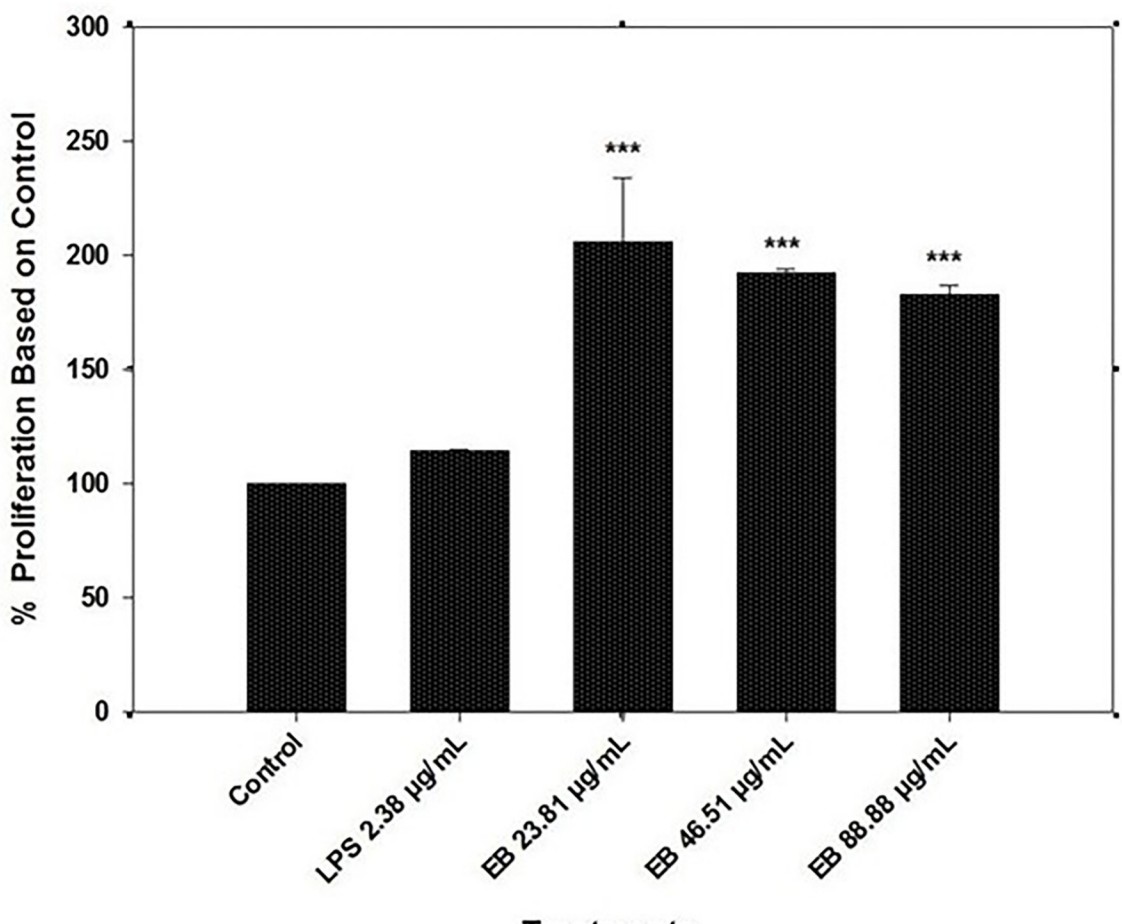

**Fig 3. Proliferation of fish spleen B-cells in the prescence of LPS and various concentrations of elderberry extracts (23.81 µg/mL, 46.51 µg/mL, and 88.88 µg/mL).** All the elderberry extract concentrations were tested in the presence of LPS. Results are represented in means ± SEM. Asterisks represent a significant proliferation difference between treatment and control cell growth (*** = P<0.001).

### 3.3. Spleen B-cell proliferation assay

B-cell proliferation assay showed the potential of the 13% anthocyanin standardized elderberry extracts as an immune-stimulating compound. There was significantly higher Spleen B-cell proliferation at 23.81 µg/mL elderberry extracts concentration than the control and LPS-only treatments (2.38 µg/mL) in the presence of non-lectin mitogen/PAMPs LPS (P < 0.001) (Fig 3). The LPS-only (2.38 µg/mL) treatment was also capable of stimulating B-cells compared to the control. However, the data represents a significantly (P < 0.001) higher spleen B-cell proliferation rate at 46.51 µg/mL and 88.88 µg/mL concentrations than control, there is no indication of the effect of concentration dependency (Fig 3). At 23.81 µg/mL concentration, the proliferation was ~2.1 times higher than the control; likewise, 46.51 and 88.88 µg/mL concentrations showed an analogous proliferation rate for spleen B-cells (~1.9 times) (Fig 3).

### 3.4. Quantification of IL-2

The quantification result of the IL-2 showed a significantly higher IL-2 concentration than control cells in different elderberry concentrations (23.81, 46.51, and 88.88 µg/mL) in the

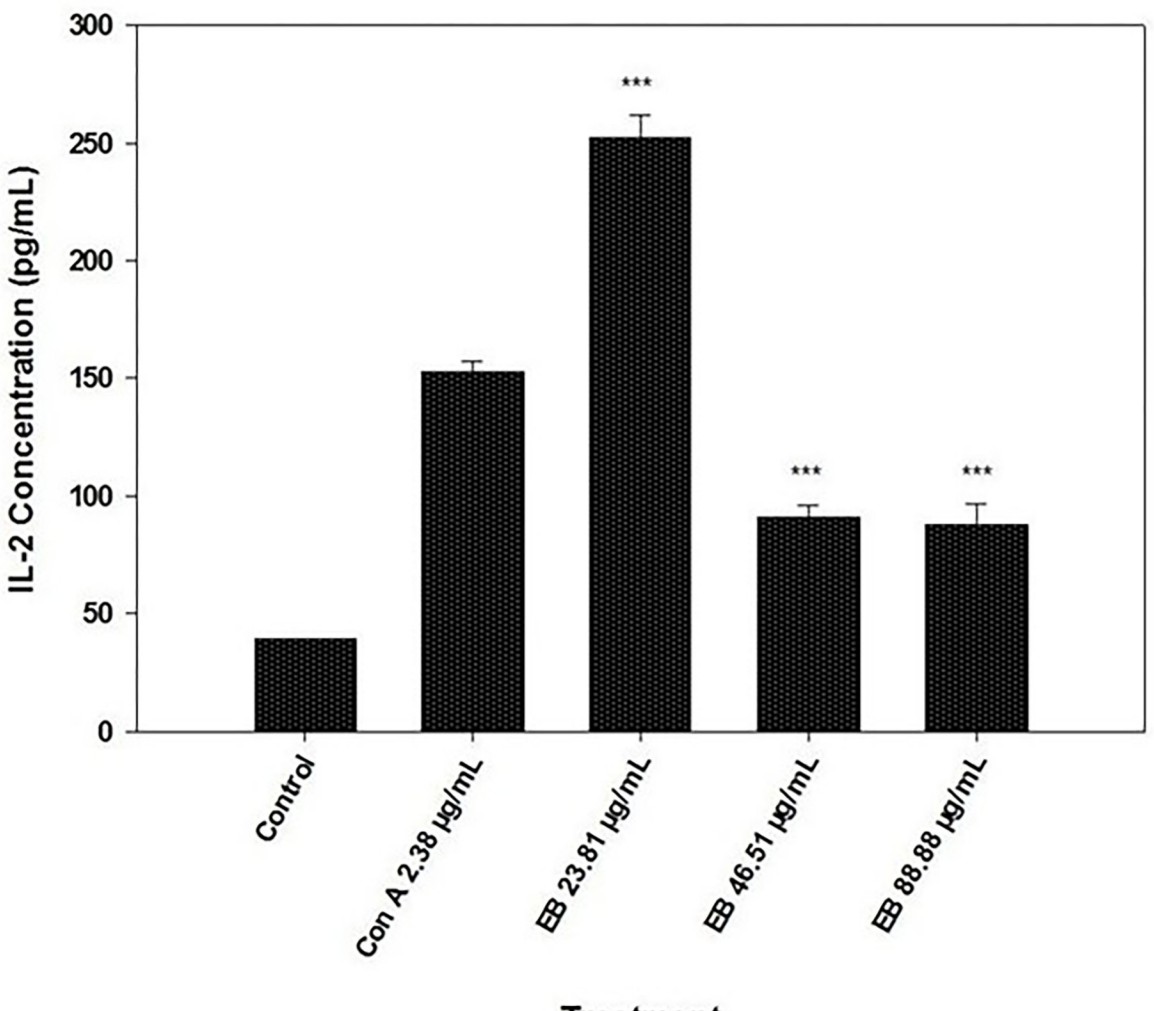

**Fig 4. Quantification of fish IL-2 concentration in the prescence of Con A and various concentrations of elderberry extracts (23.81 μg/mL, 46.51 μg/mL, and 88.88 μg/mL).** All the elderberry extract concetrations were tested in the presence of Con A. Results are represented in means ± SEM. Asterisks represent a significant proliferation difference between treatment and control cell IL-2 production (*** = P<0.001).

presence of Con A (2.38 μg/mL) (P < 0.001) (Fig 4). However, the data showed significantly higher IL-2 concentration only at 23.81 μg/mL than Con A only treatment (2.38 μg/mL) (P < 0.001) (Fig 4). Although 46.51 μg/mL and 88.88 μg/mL demonstrated significantly higher concentrations than the control (P < 0.001), the IL-2 concentrations of these two elderberry extract concentrations (46.51 μg/mL and 88.88 μg/mL) are significantly lower than Con A only treatment (2.38 μg/mL) (P < 0.001) (Fig 4). Furthermore, there was no dose-dependent increase in the IL-2 level observed at a higher dose (Fig 4).

## 4. Discussion

Black elderberry (EB) (*Sambucus nigra*) extract has been evident as a potent immune-modulating agent in several studies to boost immunity against protozoa parasites, gram-positive bacteria, gram-negative bacteria, and viruses [19, 51–55]. Several studies showed the major potent

immune-modulating components of the EB extract are anthocyanins, phenolic acids, and flavonoids [40, 56]. Although the immune-stimulating role of these secondary metabolites is well established in the literature, immunological studies relating to fish immune cells are limited to our knowledge [14, 37, 57–59]. In addition, most of the experiments performed to understand the role of the nutraceuticals were performed in the *in vivo* condition, which does not provide specific information regarding the capabilities of the nutraceuticals at the cellular level. Therefore, we examined the role of EB as an immune-stimulating factor in T- and B-cell proliferation and IL-2 production *in-vitro* in the spleen and thymus of a fish model; we used the most cultivated fish species, Nile tilapia for our study [60].

To determine the effect of the EB extracts, we used spleen and thymus T-cells as they engage in a crucial role in the cell-mediated and humoral immune response [21, 32]. In the *in-vitro* study, we observed an increased T-cell proliferation for all three concentrations of EB extract in the presence of co-stimulator mitogen ConA, and the response was not dose-dependent. In contrast, in the case of Thymus T-cell proliferation, a dose-dependent effect of the EB extract was observed. Also, the thymus T-cells required a higher dose (46.51 and 88.88 µg/mL) of the EB extract to provide a significant proliferation difference than the control. The EB extracts' variable response might be because, spleen contains mature T-cells, whereas the thymus contains mostly immature T-cells [61, 62]. As the impact of EB extracts on fish T-cells was not reported previously, we are unable to find homologous data to compare with; however, the data showed EB extracts heightened the proliferation of rat L-6 myoblasts by 242% [63]. Additionally, in humans, elderberry extract has been reported to have increased Th2 immune response at 72 µg/mL concentration [19]. The murine model exhibited black elderberry supplement increased T helper cells responses, which eventually leads to protection against leishmaniasis [52]. Taken together, based on our results and previous studies, EB extracts are either able to perform as a direct immunostimulant to T-cells, or it could also play a role as an immunomodulator through T-cells activation.

Although there is an intimate connection between cell-mediated and humoral immune responses [21], we attempted to examine the role of EB on humoral immunity by examining the spleen B-cell proliferation rate. Our data showed that EB extract could increase B-cell proliferation in the presence of gram-negative bacterial PAMPs or non-lectin mitogen LPS. All three EB concentrations showed significantly higher proliferation than the control. Earlier evaluation of the peptide extract from elderberry flowers also demonstrates antimicrobial activity against various aquaculture-related gram-negative bacterial pathogens such as *Escherichia coli*, *Vibrio anguillarum*, *Vibrio ordalii*, *Flavobacterium psychrophilum*, and *Aeromonas salmonicida*, which supports our results [64]. Also, the elderberry and its flower extracts exhibit strong complement fixating and inhibitory activity on nitric oxide (NO) production in LPS-activated RAW 264.7 macrophages and dendritic cells, which have therapeutic value in inflammatory diseases [65]. We do not have the privilege to directly compare our B-cell proliferation study with other reports; however, the previous cell line studies demonstrated cyanidin glycoside-rich extracts stimulated the proliferation of IPEC-1 cells, therefore, preventing oxidative stress-associated diseases [43]. The results presented here along with previous studies suggest that EB extract has the ability to enhance not only the cellular but also the humoral immune response.

In addition to the proliferation of the T- and B-cells, we evaluated the role of EB extract on the production of IL-2, which is mainly produced by T helper cells in response to immune stimulators such as ConA. Our data suggest that EB can have an IL-2 increasing effect on the T-cells at 23.81 µg/mL concentration of EB in the presence of ConA. In contrast, the 46.51 and 88.88 µg/mL concentration of EB extract significantly decreased the production of IL-2 even lower than the ConA only treatment (Fig 4). This inhibitory effect of higher concentrations is

synergistic with the finding of Schon et al., 2021 where the 72 μg/mL of EB extract showed significantly less (13%) IL-2 production than ConA/SEB only treatment [19]. Furthermore, it is worth mentioning that Schon et al., 2021 found equivalent IL-2 levels at 18 and 36 μg/mL compared to ConA/SEB-only treatments. It is difficult to determine whether higher-level IL-2 is good or bad for health and is dependent on the intended effect as IL-2 has an immunoregulatory role [36]. IL-2 promotes inflammatory responses through the generation of T-helper1 and T-helper2 effector cells [35]. On the contrary, IL-2 also blocks the differentiation of T cells into Th17 effectors and promotes the development or maintenance of peripheral T-regs [35]. Also, elderberry extracts have been mentioned to be anti-inflammatory through the reduction of pro-inflammatory cytokines TNF-alpha, IFN-gamma, and IL-2 [19, 66]. Therefore, 13% anthocyanin standardized elderberry extracts might be eligible as an immune modulator based on the circumstances of present or evolving inflammation [19, 35, 36, 66].

## 5. Conclusions

The effect of the elderberry extract was examined *in-vitro* as an immune-stimulating agent. In the *in-vitro* study, elderberry extract showed significantly higher clonal proliferation of T-cells for both the spleen and thymus. The B-cell proliferation also displayed the potential of the elderberry extract as an immune system stimulant. The cytokine IL-2 immunoassay demonstrated the elderberry extract's capability as an immunomodulator, which can regulate both cellular and humoral defense responses. With additional research, elderberry extracts could be used as a supplement in the aquaculture feed stimulate immune responses in fish [67]. The findings may also have applications in higher vertebrates such as humans.

## Supporting information

**S1 Dataset. Minimal dataset.**
(XLSX)

## Acknowledgments

We acknowledge the contribution of Dr. Elliott Blumenthal, Associate Professor, Department of Biology, Purdue University Fort Wayne, Indiana, USA, for his help during the experiment.

## Author Contributions

**Conceptualization:** Ahmed Mustafa.

**Data curation:** Md Mursalin Khan.

**Formal analysis:** Md Mursalin Khan.

**Investigation:** Md Mursalin Khan, Asif Mortuza, Md Ibrahim, Ahmed Mustafa.

**Methodology:** Ahmed Mustafa.

**Project administration:** Ahmed Mustafa.

**Resources:** Ahmed Mustafa.

**Supervision:** Ahmed Mustafa.

**Validation:** Asif Mortuza, Md Ibrahim.

**Writing – original draft:** Md Mursalin Khan.

**Writing – review & editing:** Asif Mortuza, Md Ibrahim, Ahmed Mustafa.

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
