## [Decision Letter · Decision Letter 0]

29 Nov 2022

PONE-D-22-27453

Assessment of the role of anthocyanin standardized elderberry (Sambucus nigra) extract as an immune-stimulating nutraceutical of Nile tilapia, Oreochromis niloticus

PLOS ONE

Dear Dr. Mustafa,

Thank you for submitting your manuscript to PLOS ONE. After careful consideration, we feel that it has merit but does not fully meet PLOS ONE’s publication criteria as it currently stands. Therefore, we invite you to submit a revised version of the manuscript that addresses the points raised during the review process.

We look forward to receiving your revised manuscript.

Kind regards,

C. Michael Greenlief, Ph.D.

Academic Editor

PLOS ONE

Journal Requirements:

Additional Editor Comments:

The manuscript is well-written and the discussion is supported by the data provided. Please revise the manuscript taking into account the reviewers' comments.

Reviewers' comments:

Reviewer's Responses to Questions

**Comments to the Author**

1. Is the manuscript technically sound, and do the data support the conclusions?

Reviewer #1: Yes

Reviewer #2: Yes

2. Has the statistical analysis been performed appropriately and rigorously? 

Reviewer #1: Yes

Reviewer #2: Yes

3. Have the authors made all data underlying the findings in their manuscript fully available?

Reviewer #1: Yes

Reviewer #2: Yes

4. Is the manuscript presented in an intelligible fashion and written in standard English?

Reviewer #1: Yes

Reviewer #2: Yes

5. Review Comments to the Author

Reviewer #1: The topic of scientific research is relevant and promising. The study of nutraceuticals and their relationship with immunity is an expanding area of research. In their work, the authors of a scientific publication focused on the immunological effect of a membrane-separated 13% anthocyanin standardized elderberry extract with the excipient maltodextrin, a widely used first-line nutraceutical for improving immunity, in aquaculture fish, Nile tilapia. The study fully evaluated the potential of elderberry extract that the authors showed to a sufficient extent using the obtained data of the ability to enhance the proliferation of lymphocytes and the production of interleukin-2 in vitro using spleen and thymus lymphocytes.

The work was carried out at a high scientific level.

The results of the obtained scientific experiments can be used to conduct further research on the use of elderberry extract as an additive to commercial feed in aquaculture to reduce stress and stimulate the immune response in fish. The results of these studies should be used for further scientific experiments, to confirm the possibility of a more extended application.

Reviewer #2: This is an interesting study using a standardized elderberry extract. Overall the studies are described well and are appropriate. There are a few minor concerns that the authors should address and these are listed below.

1. Introduction, paragraph 1: The author should state that the elderberries used in the study are a European variety instead of leaving it for the experimental section.

2. Methods and Materials section, 2.2: The authors should describe (or provide references) for the methods used to characterize the elderberry extract. A general reader will have difficulty understanding this section as written.

6. PLOS authors have the option to publish the peer review history of their article (what does this mean?). If published, this will include your full peer review and any attached files.

Reviewer #1: **Yes: **Leonid Burak

Reviewer #2: No

---

## [Author Response · Author response to Decision Letter 0]

6 Dec 2022

Ref: PONE-D-22-27453

RESPONSE TO REVIEWERS

We would like to thank the editor and the reviewers for their kind considerations and suggestions to improve the manuscript. 

Following the suggestion of the reviewers/editor:

The manuscript has been edited to meet the criteria and formatting of PLOS ONE. The methods section was edited, and the manuscript now contains a separate ethics statement including the name of the ethics body and the approved protocol number. The references were edited to meet the journal formatting requirements. General grammatical errors and spellings have been corrected. 

Response to reviewers

To Reviewer #1: Thank you for reading our work! We appreciate your feedback. 

To Reviewer #2: Thank you for your feedback! Following your suggestions, we have now edited the introduction to include the information that the Elderberry under study is of the Austrian variety. The methods section was edited to include further details of the extraction process by the manufacturer to clarify the description of the elderberry extract as stated in the literature accompanying the product. 

We believe these changes will make the manuscript better and we are thankful for the suggestions. Please let us know if there are any other suggestions, improvements to be made or we missed.

---

## [Editor Report · Decision Letter 1]

7 Dec 2022

Assessment of the role of anthocyanin standardized elderberry (Sambucus nigra) extract as an immune-stimulating nutraceutical of Nile tilapia, Oreochromis niloticus

PONE-D-22-27453R1

Dear Dr. Mustafa,

We’re pleased to inform you that your manuscript has been judged scientifically suitable for publication and will be formally accepted for publication once it meets all outstanding technical requirements.

Kind regards,

C. Michael Greenlief, Ph.D.

Academic Editor

PLOS ONE

Additional Editor Comments (optional):

The authors have addressed well the minor concerns raised by the reviewers.
---

## [Editor Report · Acceptance letter]

22 Dec 2022

PONE-D-22-27453R1 

Assessment of the role of anthocyanin standardized elderberry (*Sambucus nigra*) extract as an immune-stimulating nutraceutical of Nile tilapia, *Oreochromis niloticus*

Dear Dr. Mustafa:

I'm pleased to inform you that your manuscript has been deemed suitable for publication in PLOS ONE. Congratulations! Your manuscript is now with our production department. 

Kind regards, 

on behalf of

Dr. Charles Michael Greenlief 

Academic Editor

PLOS ONE